

# Functional genetic variants in complement component 7 confer susceptibility to gastric cancer

Siyue Wang[1,2], Wenqian Hu[1], Yuning Xie[1], Hongjiao Wu[1], Zhenxian Jia[1], Zhi Zhang[3] and Xuemei Zhang[1,2]

[1] School of Public Health, North China University of Science and Technology, Tangshan, China
[2] College of Life Science, North China University of Science and Technology, Tangshan, China
[3] Affiliated Tangshan Gongren Hospital, North China University of Science and Technology, Tangshan, China

## ABSTRACT

**Background**. Complement system plays an important role in innate immunity which involved in the changes tumor immune microenvironment by mediating the inflammatory response. This study aims to explore the relationship between complement component 7 (*C7*) polymorphisms and the risk of gastric cancer (GC).

**Materials and Methods**. All selected SNPs of *C7* were genotyped in 471 patients and 471 controls using the polymerase chain reaction-restriction fragment length polymorphism (PCR-RFLP). Odds ratios (ORs) and 95% confidence intervals (*CI*s) were calculated by unconditional Logistic regression to analyze the relationship between each genotype and the genetic susceptibility to gastric cancer. The level of *C7* expression in GC was analyzed by Gene Expression Profiling Interactive Analysis (GEPIA) and detected by Enzyme Linked Immunosorbent Assay. Kaplan–Meier plotter were used to reveal *C7* of prognostic value in GC. We examined SNPs associated with the expression of *C7* using the GTEx database. The effect of *C7* polymorphisms on the regulatory activity of *C7* was detected by luciferase reporter assay.

**Results**. Unconditional logistic regression showed that individuals with *C7* rs1376178 AA or CA genotype had a higher risk of GC with OR (95% CI) of 2.09 (1.43–3.03) and 1.88 (1.35–2.63), respectively. For *C7* rs1061429 C > A polymorphism, AA genotype was associated with the elevated risk for developing gastric cancer (OR = 2.16, 95% CI [1.37–3.38]). In stratified analysis, *C7* rs1376178 AA genotype increased the risk of GC among males (OR = 2.88, 95% CI [1.81–4.58]), but not among females (OR = 1.06, 95% CI [0.55–2.06]). Individuals carrying rs1061429 AA significantly increased the risk of gastric cancer among youngers (OR = 2.84, 95% CI [1.39–5.80]) and non-smokers (OR = 2.79, 95% CI [1.63–4.77]). *C7* was overexpressed in gastric cancer tissues and serum of cancer patients and was significantly associated with the prognosis. *C7* rs1061429 C > A variant contributed to reduced protein level of *C7* (*P* = 0.029), but rs1376178 didn't. Luciferase reporter assay showed that rs1376178C-containing plasmid exhibited 2.86-fold higher luciferase activity than rs1376178 A-containing plasmid (*P* < 0.001). We also found that rs1061429A allele contributed 1.34-fold increased luciferase activity than rs1061429C allele when co-transfected with miR-591 (*P* = 0.0012).

**Conclusions**. These findings highlight the role of *C7* in the development of gastric cancer.

Corresponding author
Xuemei Zhang, jyxuemei@gmail.com

## INTRODUCTION

Gastric cancer is one of the common malignant cancer worldwide, especially in East Asia (*Siegel et al., 2021*). The incidence of gastric cancer ranks the third and fourth among men and women in China, respectively (*Chen et al., 2016*). For most gastric cancer patients, more symptoms are usually considered to be related to an advanced stage and the surgical resection is still the main therapeutic choice (*Digklia & Wagner, 2016*; *Song et al., 2017*). Several genome-wide association studies (GWAS) datasets showed that genetic variants were significantly associated with gastric cancer risk (*Jin et al., 2020*; *Saeki et al., 2013*).

The human immune system is made up of two distinct parts, innate immune system and adaptive immune system. As the body's first line of defense against germs and foreign substances, innate immune system provides immediate and non-specific immune responses, which is different from the way in which adaptive immune system specifically recognizes and eliminates pathogens through specialized T and B lymphocytes (*Berraondo et al., 2016*; *Saeki et al., 2013*). Complement system is a critical component of innate immunity, which can be activated by three major pathways: the classical pathway, the alternative pathway, and the Mannose-binding Lectin (MBL) pathway. Many studies have demonstrated that complement activation enhances innate immunity against cancer through immune infiltrating or complement-dependent cytotoxicity (*Bao et al., 2020*; *Park et al., 2012*). In addition, complement components *C5a* and *C3a* generated by complement cascades facilitate cancer cell proliferation and regeneration (*Markiewski et al., 2008*; *Ostrand-Rosenberg, 2008*).

Complement component 7 (*C7*) plays a central role in the activation of complement system as the final product of the complement cascade (*Würzner, 2000*), which acts as one of major rate-limiting factors for the formation of membrane attack complex (MAC) (*Walport, 2001*; *Ying et al., 2016*). A study showed that *C7* had increased expressed in liver cancer stem cells and enhanced the stemness of liver cancer cells by up-regulating Nanog, Oct4, Sox2, and C-myc (*Seol et al., 2016*). *C7* is also identified as a potential tumor suppressor and may serve as a prognostic biomarker for certain cancers (*Chen et al., 2020*; *Ying et al., 2016*).

Complement gene polymorphism is closely related to the occurrence of cancer. DAF (decay accelerating factor), as one of key inhibitors of the complement system, inhibits complement activation by preventing the formation of C3/C5 convertase from interfering with the formation of MAC (*Mikesch et al., 2006*; *Spendlove et al., 2006*). Studies have indicated that DAF rs2564978 T > C variant contributed to an increased risk of NSCLC (*Zhang et al., 2017*) and rs10746463 G > A polymorphism was related to elevated risk of gastric cancer (*Song et al., 2015*). Complement receptor 1 (CR1), acting as the receptor of C3b and C4b to inhibit the complement activity (*Liu & Niu, 2009*). The tag genetic variant rs9429942 in CR1 had great effect on the susceptibility to gastric cancer (*Zhao et al., 2015*).

However, the relationship between the polymorphisms of *C7* and the susceptibility to gastric cancer still needs to be explored.

In this study, we conducted a case-control study in the Chinese population to verify the hypothesis that the potential functional polymorphism of *C7* contributes to the susceptibility of gastric cancer.

## MATERIALS AND METHODS

### Study population

This case-control study contains 471 gastric cancer patients and 471 healthy controls. All patients were recruited from North China University of Science and Technology Affiliated Tangshan Renmin Hospital and Affiliated Tangshan Gongren Hospital from January 2011 to May 2015. Healthy individuals were from a large population underwent physical examinations in same area during the same period. We also detected the *C7* expression level in serum of 70 healthy control and 70 gastric cancer patients. This study was approved by Institutional Review Board of North China University of Science and Technology (2019021). All subjects signed an informed consent form.

### Potentially functional SNPs filtering

Based on NCBI dbSNP (https://www.ncbi.nlm.nih.gov/snp/) and Ensembl (http://asia.ensembl.org/index.html) database, we screened out all SNPs in the promoter (2,000 bp upstream of the transcription revelation site) and 3′ untranslated region (3′UTR) of *C7* with minor allele frequency (MAF) over than 0.05 in Chinese population. Alibaba 2.1 (http://gene-regulation.com/pub/programs/alibaba2/index.html) tool was then used to predict the binding ability of transcription factors and SNP info Web server (https://manticore.niehs.nih.gov/) to predict the miRNA binding changes of SNPs in 3′UTR.

### Genotyping of genetic variants

Peripheral blood DNA was extracted by using TIANamp Blood DNA Kit (TIANGEN, Beijing, China) according to manufacturer's instructions. PCR-restriction fragment length polymorphism (PCR-RFLP) analysis were applied for genotyping. The target DNA fragment containing *C7* rs1376178 or rs1061429 was amplified with primer pairs, rs1376178 PF (5′-GCTAGAATCAATGCAAAGCTATGCG-3′)/PR (5′-TCAGATCACTGTGTTGGAAAGTT-3′) and rs1061429 PF (5′-GCTAGAATCAATGC AAAGCTATGCG-3′)/PR (5′-AAGGAAAAGCTGTCCAGTGC-3′), respectively. PCR was performed in 6 μL PCR reaction mixture with 1×Ftaq PCR Mix, 20 ng genomic DNA and 0.1 μM each primer. The thermal cycling conditions for both *C7* rs1376178 and rs1061429 variants were 3 min at 94 °C followed by 30 s at 94 °C, 30 s at 60 °C, and 30 s at 72 °C for 30 cycles, and then a final extension 3 min at 72 °C. PCR products for *C7* rs1376178 C > A (125 bp) and *C7* rs1061429 C > A (242 bp) were digested by *Hha* I (NEB, Ipswich, MA, USA) and *Nco* I (NEB, Ipswich, MA, USA) and then was separated on 3% agarose gel. For quality assurance, approximately 10% of the samples were randomly selected for re-genotyping and all results were in 100% concordance.

## Bioinformatic analysis of *C7* expression

To analyze the levels of the *C7* mRNA expression in GC, the GEPIA databases were analyzed. GEPIA (http://gepia.cancer-pku.cn/) is a comprehensive and interactive web resource for analyzing cancer data, which includes 9,736 tumors and 8,587 normal samples from Genotype-Tissue Expression (GTEx) and The Cancer Genome Atlas (TCGA) (*Tang et al., 2017*). Additionally, we also extracted eQTL data from the GTEx database, where differences in gene levels under different SNPs were examined. *P*- value with < 0.05 was considered as statistical significance.

## Enzyme linked immunosorbent assay (ELISA)

The serum samples from 70 normal individuals and 70 gastric cancer patients were used to detect the level of *C7* protein by using enzyme-linked immunosorbent assay (ELISA). Human Complement (*C7*) kit were purchased from Cusabio company (Wuhan, China). We conducted the *C7* expression analysis in serum according to manufactory's instructor.

## Cell culture

Gastric cancer cell line (BGC823) was purchased from Cell Bank of Type Culture Collection of the Chinese Academy of Sciences Shanghai Institute of Biochemistry and Cell Biology. BGC823 cells were cultured in RPMI-1640 (Thermo Fisher Scientific, Waltham, MA, USA) supplemented with 10% fetal bovine serum (FBS; Thermo Fisher Scientific, Waltham, MA, USA) and 1% antibiotics (100 U/ml penicillin and 100 $\mu$g/mL streptomycin) in an atmosphere of 5% CO2 at 37 °C.

## Plasmid constructure and luciferase reporter gene assay

A 1,615 bp DNA fragment containing rs1376178 site in the promoter of *C7* was inserted into pGL3-basic plasmid (Promega, Madison, WI, USA) to conduct luciferase reporter gene assay. The PCR primers with *Kpn* I and *Xho* I (NEB, Ipswich, USA) cutting site adaptor were 5′-GGGTACCCTTTCCCACTTCCAGTGGTGC-3′) and 5′-CCGCTCGAG CTGAGATTTAGCTCCTACCCC-3′. The final plasmids with rs1376178 C or A allele were designed as pGL3$_{rs1376178C}$ and pGL3$_{rs1376178A}$, respectively. A 1,268 bp fragment with rs1061429 site in the 3′ untranslated region (UTR) was cloned into psiCHECK2 plasmid (Promega, Madison, WI, USA). The PCR primer pairs with *Xho* I and *Not* I (NEB, Ipswich, USA) sites were 5′-CCGCTCGAGTGCGAGGA AGAAGGGTTT-3′ and 5′-GAATGCGGCCGCTGGGACTGTATCCACAGAA-3′. The constructors with rs1061429C or A allele were named as psiCHECK2$_{rs1061429C}$ and psiCHECK2$_{rs1061429A}$, respectively.

BGC823 gastric cancer cells were seeded in 24-well plates at a density of $2 \times 10^5$ cells/well. As the cells reached 60–70% confluent, 300 ng of each pGL3-containing plasmids and 5ng pRL-SV40 (Promega, Madison, WI, USA) were co-transfected into the cells using Lipofectamine 2000 (Invitrogen, Carlsbad, CA, USA). For psiCHECK2-containing plasmids, 20 ng of plasmids was transfected into gastric cancer cells with or without 30pmol of miR-591 mimic (GenePharma, Shanghai, China). Gastric cancer cells were then collected 24 h after transfection. The activity of luciferase and renilla reporter gene activity were measured using GloMax 20/20 Luminometer (Promega, Madison, WI, USA).

**Table 1  Frequency distribution of the study population.**

| Variables | Cases ($n = 471$) | | Controls ($n = 471$) | | P value[a] |
|---|---|---|---|---|---|
| | NO | % | NO | % | |
| Gender | | | | | |
| Male | 332 | 70.5 | 321 | 68.2 | 0.44 |
| Female | 139 | 29.5 | 150 | 31.8 | |
| Age | | | | | |
| <60 | 209 | 44.4 | 209 | 44.4 | 1.00 |
| ≥60 | 262 | 55.6 | 262 | 55.6 | |
| Smoking status | | | | | |
| Non-smoker | 323 | 68.6 | 337 | 71.5 | 0.32 |
| smoker | 148 | 31.4 | 134 | 28.5 | |
| Drinking status | | | | | |
| No | 374 | 79.4 | 388 | 82.4 | 0.25 |
| Yes | 97 | 20.6 | 83 | 17.6 | |

**Notes.**
[a]Two-sided $\chi 2$ test.

## Statistical analysis

The statistical analyses in our study were conducted by SPSS 23.0 (SPSS Inc., Chicago, IL, USA). The Hardy-Weinberg equilibrium (HWE) of *C7* polymorphisms in controls were assessed by $\chi 2$ test. Differences of basic characteristics between cases and control subjects were evaluated by $\chi 2$ test. The association of *C7* genotypes with the susceptibility to gastric cancer was evaluated by unconditional logistic regression with OR (95% *CI*) after adjusted by age, sex, smoking status and drinking status. The interaction between gene and environment was analyzed by epiR program in R platform (version 3.6.1). We defined smokers as they smoked more than 100 cigarettes in their lifetime. Drinkers were categorized as the individuals took at least 12 drinks on one occasion during the previous year according to international guide for monitoring alcohol consumption and harm of WHO. *P* value < 0.05 was regarded as statistical significance. To explore the associations of *C7* expression level and the prognosis of gastric cancer, we performed survival analysis by Kaplan–Meier online program (https://kmplot.com/).

## RESULTS

### Subject characteristics

The distribution of select characteristics of all subjects were shown in Table 1. The study involved 471 gastric cancer patients and 471 healthy controls. There was no statistically significant difference in distribution of age and gender between cases and controls (age: $P = 1.00$; gender $P = 0.44$). Regard to the distribution of smoking and drinking status, there was no significant difference between cases and controls (smoking: $P = 0.32$; drinking $P = 0.25$).

**Table 2  Single nucleotide polymorphism information and Hardy–Weinberg test.**

| Gene | Position | SNP | Region | Allele gene | MAF | Functional change | P value |
|---|---|---|---|---|---|---|---|
| C7 | chr5:40908799 | rs1376178 | promoter | C/A | 0.45 | STAT1[a] | 0.393 |
| C7 | chr5:40981587 | rs1061429 | 3′UTR | C/A | 0.39 | has-miR-591[b] | 0.710 |

Notes.
[a] Typical transcription factor changes of promoter.
[b] Typical miRNA binding changes of 3′UTR.

**Table 3  Genotype frequencies of C7 and their association with gastric carcinoma.**

| Genotypes | Controls($n = 471$) | | Cases($n = 471$) | | OR (95% CI)[a] | P value |
|---|---|---|---|---|---|---|
| | N | (%) | N | (%) | | |
| *C7* rs1376178 | | | | | | |
| CC | 130 | 27.6 | 77 | 16.4 | 1.00ref | |
| CA | 226 | 48.0 | 252 | 53.5 | 1.88 (1.35–2.63) | <0.001 |
| AA | 115 | 24.4 | 142 | 30.1 | 2.09 (1.43–3.03) | <0.001 |
| *C7* rs1061429 | | | | | | |
| CC | 246 | 52.2 | 225 | 47.8 | 1.00ref | |
| CA | 191 | 40.6 | 179 | 38.0 | 1.03 (0.78–1.35) | 0.861 |
| AA | 34 | 7.2 | 67 | 14.2 | 2.16 (1.37–3.38) | 0.001 |

Notes.
[a] Adjusted for age, gender smoking status and drinking status.

## Association of *C7* gene polymorphisms with gastric cancer risk

After predicting potential regulatory functional SNPs, we found that *C7* rs1376178 C > A variant enhanced the binding capability to transcription factor STAT1 and *C7* rs1061429 C > A allele created a binding site with has-miR-591 in 3′ UTR. Genotype distributions of rs1376178 and rs1061429 polymorphisms in controls were conformed to Hardy-Weinberg equilibrium (HWE) (Table 2).

The relationship between each genetic variant and the susceptibility to gastric cancer was shown in Table 3. After adjusted by gender, age, drinking and smoking status, non-conditional logistic regression analysis showed that the distribution of *C7* rs1376178 AA and CA genotypes was statistically different between cases and controls ($P < 0.001$). The individuals with *C7* rs1376178 AA or CA genotype had a higher risk of GC with OR (95% CI) of 2.09 (1.43–3.03) and 1.88 (1.35–2.63), respectively. For *C7* rs1061429 C > A polymorphism, AA genotype was associated with the elevated risk for developing gastric cancer (OR = 2.16, 95% CI [1.37–3.38]).

## Stratification analysis of *C7* variants with GC risk

To further analyze the effect of age, gender, smoking and drinking on the association of *C7* variants (rs1376178 and rs1061429) with the risk of GC, we performed stratification analysis (Table 4). Gender stratification analysis showed that *C7* rs1376178 AA genotype was associated with the elevated risk of gastric cancer among males (OR = 2.88, 95% CI [1.81–4.58], $P < 0.001$), but not among females (OR = 1.06, 95% CI [0.55–2.06], $P > 0.05$). When stratified by age, individuals with *C7* rs1376178 AA had an increased risk of gastric cancer in both groups with OR (95% CI) of 2.19 (1.26–3.78) for the youngers

**Table 4** Association of *C7* rs1376178 C > A polymorphism with GC risk stratified by selected variables.

| Variables | Genotypes (Controls/Cases) | | | AA/CC model | P value |
|---|---|---|---|---|---|
| | CC | CA | AA | OR (95% CI)[a] | |
| Gender | | | | | |
| Male | 91/48 | 158/176 | 72/108 | 2.88 (1.81–4.58) | <0.001 |
| Female | 39/29 | 68/76 | 43/34 | 1.06 (0.55–2.06) | 0.862 |
| Age | | | | | |
| <60 | 66/40 | 97/108 | 46/61 | 2.19 (1.26–3.78) | 0.005 |
| ≥60 | 64/37 | 129/144 | 69/81 | 2.07 (1.23–3.48) | 0.006 |
| Smoking status | | | | | |
| No | 92/51 | 156/172 | 89/100 | 2.05 (1.31–3.21) | 0.002 |
| Yes | 38/26 | 70/80 | 26/42 | 2.40 (1.19–4.87) | 0.015 |
| Drinking status | | | | | |
| No | 105/62 | 181/196 | 102/116 | 1.94 (1.28–2.93) | 0.002 |
| Yes | 25/15 | 45/56 | 13/26 | 3.55 (1.37–9.17) | 0.009 |

**Notes.**

[a] Data were calculated by unconditional logistic regression and adjusted for age, gender, smoking and drinking status, where they were appropriate.

and 2.07 (1.23–3.48) for the elders. Our data also showed that the risk of gastric cancer was associated with the rs1376178 AA regardless of smoking and drinking status (OR = 2.05, 95% CI [1.31-3.21], $P = 0.002$ for nonsmokers; OR = 2.40, 95% CI [1.19–4.87], $P = 0.015$ for smokers; OR = 1.94, 95% CI [1.28–2.93], $P = 0.002$ for nondrinkers; OR = 3.55, 95% CI [1.37–9.17], $P = 0.009$ for drinkers).

Stratification analysis of *C7* rs1061429 polymorphism was showed in Table 5. Our data suggested that individuals with rs1061429 AA had an increased risk of gastric cancer among youngers (OR = 2.84, 95% CI [1.39–5.80], $P = 0.004$) and non-smokers (OR = 2.79, 95% CI [1.63–4.77], $P < 0.001$), but not among elders (OR = 1.81, 95% CI [1.00–3.28], $P = 0.050$) and smokers (OR = 1.03, 95% CI [0.43–2.5], $P = 0.947$). The stratification analysis by gender or drinking status showed that rs1061429 AA genotype was contributed to the risk of gastric cancer regardless of gender and drinking status (OR = 1.94, 95% CI [1.13–3.33], $P = 0.017$ for males; OR = 2.87, 95% CI [1.26–6.54], $P = 0.012$ for females; OR = 1.92, 95% CI [1.18–3.13], $P = 0.008$ for nondrinkers; OR = 5.23, 95% CI [1.34–20.40], $P = 0.017$ for drinkers). Use R package epiR to build Logistic regression gene-environment interaction mode, we found that rs1061429 had no additive and multiplicative interaction with drinking or gender to affect the risk of gastric cancer ($P > 0.05$).

## The expression of *C7* in gastric cancer and its effect on prognosis and clinical-pathological characteristics of gastric cancer

Furthermore, we explore the potential function of gene *C7* in gastric cancer, based on GEPIA data, we analyzed the differential expression of *C7* in gastric cancer tissues and adjacent normal tissues and found that the level of *C7* mRNA in gastric cancer tissues ($n = 408$) was significantly depressed when compared with that in adjacent normal tissues

**Table 5 Association of *C7* rs1061429 C > A polymorphism with GC risk stratified by selected variables.**

| Variables | Genotypes (Controls/Cases) | | | AA/CC model | P value |
|---|---|---|---|---|---|
| | CC | CA | AA | OR (95% CI)[a] | |
| Gender | | | | | |
| Male | 172/165 | 125/123 | 24/44 | 1.94 (1.13–3.33) | 0.017 |
| Female | 74/60 | 66/56 | 10/23 | 2.87 (1.26–6.54) | 0.012 |
| Age | | | | | |
| <60 | 110/106 | 87/70 | 12/33 | 2.84 (1.39–5.80) | 0.004 |
| ≥60 | 136/119 | 104/109 | 22/34 | 1.81 (1.00–3.28) | 0.050 |
| Smoking status | | | | | |
| Non-smoker | 172/149 | 142/119 | 23/55 | 2.79 (1.63–4.77) | <0.001 |
| smoker | 74/76 | 49/60 | 11/12 | 1.03 (0.43–2.5) | 0.947 |
| Drinking status | | | | | |
| No | 203/180 | 154/141 | 31/53 | 1.92 (1.18–3.13) | 0.008 |
| Yes | 43/45 | 37/38 | 3/14 | 5.23 (1.34–20.40) | 0.017 |

**Notes.**

[a]Data were calculated by unconditional logistic regression and adjusted for age, gender, smoking and drinking status, where they were appropriate.

($n = 211$) (Fig. 1A). To verify the result from GEPIA data, we further compared the expression of *C7* in serum between normal individuals and gastric cancer patients and found a significant differential expression of *C7* ($P = 0.043$) (Fig. 1B).

We also analyzed the impact of *C7* expression on the prognosis of gastric cancer using Kaplan–Meier online program and demonstrated that higher expression of *C7* was related with poor overall survival time (OS) and post-progression survival time (PPS) of gastric patients with HR (95% CI) of 1.29 (1.09–1.53) and 1.51 (1.19–1.91), respectively (Figs. 1C and 1D). We also evaluated the association of *C7* expression level with TNM stage. Our data didn't show any correlation between *C7* expression and TNM stage (Fig. S1 ).

### *C7* genotypes and clinical-pathological characteristics

We used Kendall's rank correlation tests to estimate the correlation between these two SNPs and clinicopathological characteristics. Our results showed that rs1061029 C > A variant has a positive correlation with lymph node metastasis, but the correlation strength was low ($P = 0.022$, Kendall's Tau-b = 0.104). We didn't find any correlation between rs1376178 C > A variant and TNM stage, tumor size, lymph node metastasis, distant metastasis (Table 6).

### The effect of *C7* polymorphisms on the regulatory activity of *C7*

To substantiate the association between the identified SNPs and GC risk, using the genotype and gene expression data of 324 normal gastric tissues in the GTEx to analyze, the eQTL data results showed that the mRNA expression of *C7* was significantly related to the rs1061429 genotype ($P < 0.001$) (Fig. 2A). However, there is no eQTL data related to rs1376178 in GEPIA database.

We then measured the *C7* protein level in serum of 70 individuals. For rs1061429 variant, the expression of *C7* was significantly lower in individuals with AA genotype

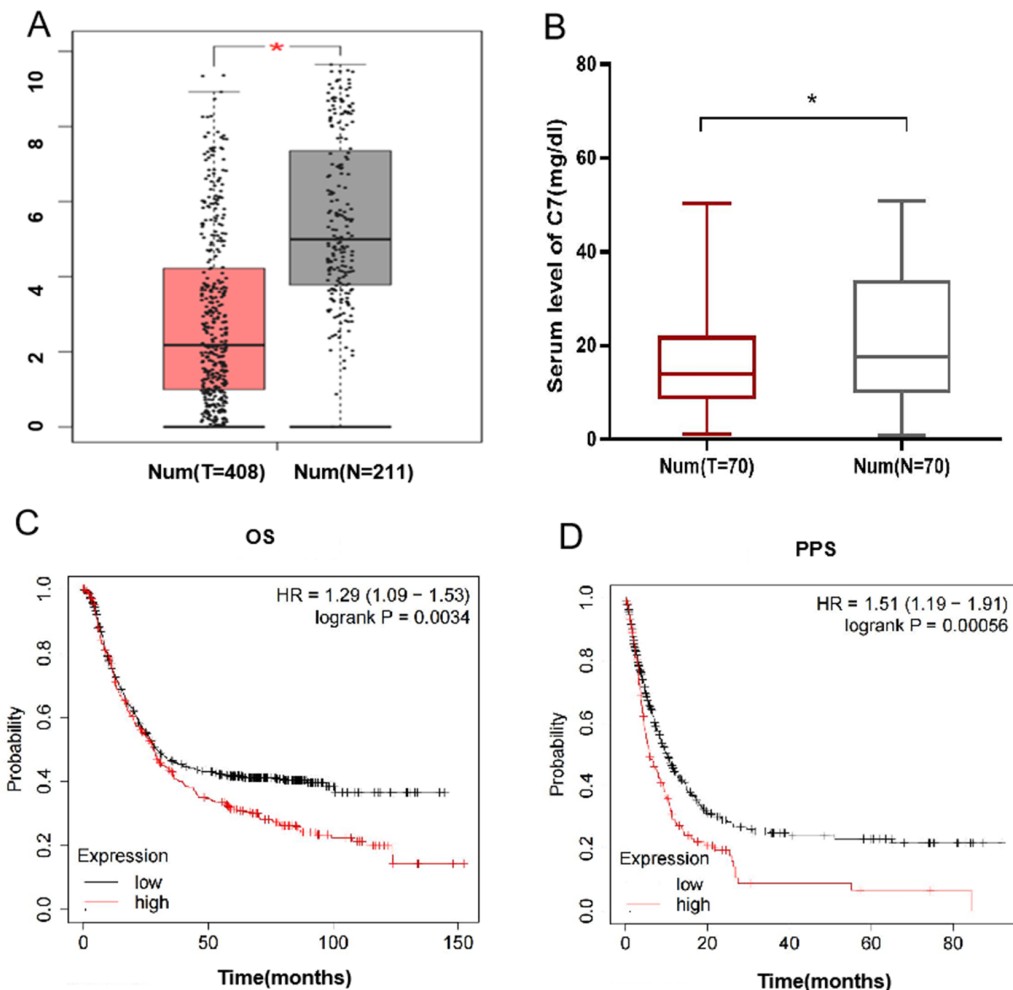

**Figure 1  C7 expression in gastric cancer and its effect on the prognosis of gastric cancer patients.** (A) C7 expression in gastric cancer and adjacent normal tissues in GEPIA database. (*$P < 0.05$). (B) *C7* expression in serum of normal individuals and gastric cancer patients. (C) Overall survival and (D) post-progression survival analysis of gastric cancer patients based on C7 expression.

**Table 6  Correlation between *C7* polymorphisms and clinical characteristics.**

| Clinical pathological characteristics | rs1061429 C > A | | rs1376178 C > A | |
|---|---|---|---|---|
| | Tau-b | *P* value | Tau-b | *P* value |
| TNM stage | 0.074 | 0.089 | 0.021 | 0.631 |
| Tumor size | 0.071 | 0.123 | −0.004 | 0.935 |
| Lymph node metastasis | 0.104 | 0.022 | 0.065 | 0.156 |
| Distant metastasis | 0.086 | 0.075 | 0.027 | 0.569 |

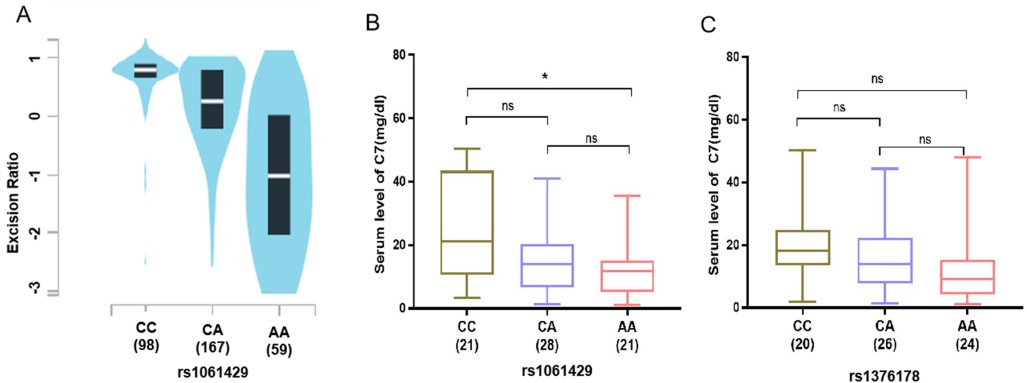

**Figure 2** **Association of C7 expression in gastric cancer patients with C7 genetic polymorphisms.** (A) rs1061429 genotypes from GTEx database. (B) rs1061429 genotypes and (C) rs1376178 genotypes in serum.

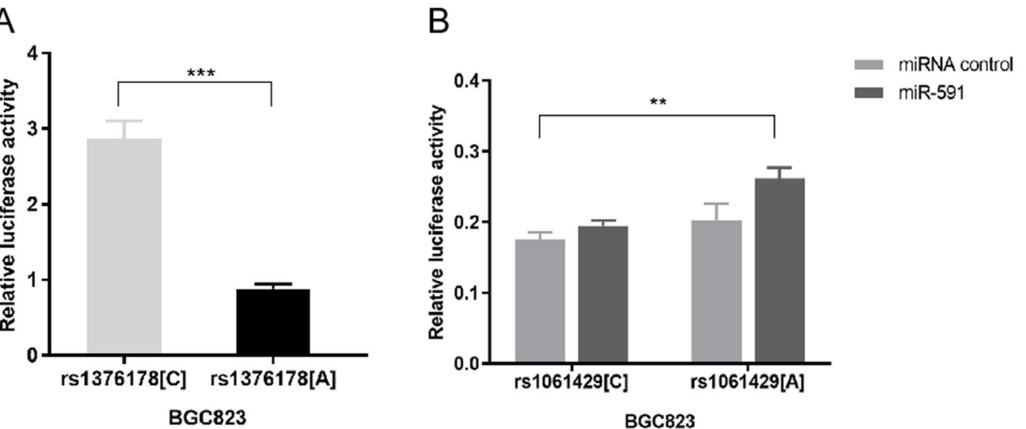

**Figure 3** **The effect of C7 polymorphisms on transcription activity.** (A) Different allele of rs1376178 had different regulatory effects in BGC823 cells. (B) Different allele of rs1061429 had different 3′UTR activity in BGC823 cells. ***$P < 0.001$; **$P < 0.01$.

when compared with those with CC genotype ($P = 0.029$) (Fig. 2B). For rs1376178 polymorphism, our data didn't show any effect on *C7* expression (Fig. 2C).

Our luciferase reporter assay showed that pGL3 $_{rs1376178C}$ exhibited 2.86-fold higher luciferase activity than pGL3$_{rs1376178A}$ ($P < 0.001$) (Fig. 3A). We also found that psiCHECK2$_{rs1061429A}$ had 1.34-fold increased luciferase activity in comparison to psiCHECK2$_{rs1061429C}$when co-transfected with miR-591 ($P = 0.0012$) (Fig. 3B). There was no effect of rs1061429 polymorphism on the reporter gene activity without additional miR-591.

## DISCUSSION

The development of gastric cancer is a long-term multistage process with complex etiology. Genetic epidemiological studies have shown that gastric cancer is the result of long-term effects of environmental and individual genetic factors (*Jin et al., 2020*). Many studies provided strong evidence for the effect of complement activation on tumor development (*Kwak et al., 2018*; *Markiewski et al., 2008*; *Roumenina et al., 2019*). *C7* is a terminal component of complement activation which plays essential roles within innate immunity (*Fujita, Matsushita & Endo, 2004*; *Walport, 2001*). Membrane-associated *C7* was acted a regulator of the excessive proinflammatory reaction (*Bossi et al., 2009*). As an essential part of the membrane attack complex (MAC), *C7* participated in various microbial defense responses and immune injury responses. The lack of *C7* may affect the function of MAC and further increase the susceptibility to infection (*Barroso et al., 2010*; *Sarma & Ward, 2011*). Tumor infection promoted cancer aggression, prevented inflammation, reduces metastasis and improved anti-tumor treatment (*Maller et al., 2021*).*C7* played an important role in the occurrence of various cancers. It was demonstrated that *C7* affected the progression of liver cancer *via* affecting the transcription of stemness factors (*Seol et al., 2016*). Similarly, *C7* was verified to be related to the prognosis of prostate cancer patients (*Chen et al., 2020*). Consistent with our finding which showed a significantly down-regulated *C7* in gastric cancer tissue, researchers also found that the *C7* was down expressed in ovarian, it was also demonstrated that the decreased expression of *C7* was related to poor differentiation in patients with NSCLC (*Ying et al., 2016*). These studies indicated its significance in the occurrence and development of tumors.

Single Nucleotide Polymorphism (SNP) was widely present in human genome and is the most common type of genetic variation, which can affect gene regulation by changing gene structure or expression (*Zhang et al., 2014*). There are few reports on the relationship between *C7* genetic variants and cancer risk. In this study, we discovered that *C7* rs1376178 C > A increased the susceptibility to gastric cancer. This was consistent with the *C7* expression analysis and dual-luciferase reporter gene results which showed that the rs1376178 A allele significantly reduced the promoter activity and the expression level of *C7*. For rs1061429 polymorphism, we found that *C7* rs1061429 C > A increased the binding ability of has-miR-591 to reduce the activity of reporter gene. Literature reported dysregulation of miR-591 confer paclitaxel resistance to ovarian cancer (*Huh et al., 2013*), and circ_0091581 could promote the progression of hepatocellular carcinoma through miR-591/FOSL2 Axis (*Ji et al., 2021*). Therefore, binding of has mir-591 may inhibit translation of *C7* and attenuate its expression further participate in the development of gastric cancer.

Gastric cancer is a complex disease, in which the interaction of genetic variants with several compounding factors, such as age, sex and environment, has been demonstrated to modulate the risk phenotypes (*Favé et al., 2018*; *Jiang, Holmes & McVean, 2021*). For gastric cancer, the environmental risk factors involved in *Helicobacter pylori* (*H. pylori*) infection, smoking and drinking (*Brenner, Rothenbacher & Arndt, 2009*; *Terry, Gaudet & Gammon, 2002*). Due to the missing data of *H. pylori* infection, we only analyzed the impact

of these compounding factors, including age, sex, smoking status and drinking status on genetic risk of disease.

Smoking was one of established and important risk factors contributing to the risk of gastric cancer (*Butt et al., 2019*). Thus, we analyzed the effects of *C7* variants on the risk of GC when stratified by smoking status. Our data presented that rs1061429 variant had an effect on gastric cancer risk among non-smokers, but not among smokers. Cigarette smoke contains multiple known human carcinogens. Exposure to nicotine-derived nitrosamine ketone, a key carcinogenic ingredient of cigarette smoke, had been proved to result in mitochondrial dysfunction (*Wu et al., 2019*), to promote immune dysfunction, and further to influence tumor immune microenvironment (*De la Iglesia et al., 2020*; *Lee, Taneja & Vassallo, 2012*). It has been reported that cigarette smoke can induce oxidative injury and dose-dependently stimulated gastric cancer cell proliferation (*Bhattacharyya et al., 2014*; *Shin et al., 2004*). Smoking status could affect the association of *C7* rs1061429 variant with the gastric cancer risk which was consistent with the report on gene-environment interaction between smoking and SNP in decay-accelerating factor (DAF) gene (*Song et al., 2015*). This could be supported by which tobacco enhanced the activation of the classical pathway of the complement system (*Yin et al., 2008*).

Drinking was another important risk factor for gastric cancer (*Ma et al., 2017*; *Na & Lee, 2017*). In current study, individuals with *C7* rs1376178AA or rs1061429AA genotype were contributed to the risk of gastric cancer regardless of drinking status. The result was consistent with some previous studies that drinking significantly effect on association between polymorphisms and gastric cancer risk (*Li et al., 2021*; *Qiu et al., 2015*).

Besides smoking and drinking, age also contributed to the occurrence and development of a variety of cancers (*Hansen et al. 2019*). In this study, when stratified by age, our data showed that individuals carrying *C7* rs1061429 AA genotype had an increased risk of gastric cancer among youngers, but not among elders. The pathogenesis of gastric cancer in the elders was mostly induced by environmental factors (*Forman & Burley, 2006*); however, young patients are more effected by genetic factors (*Machlowska et al., 2020*). Several previous studies from our laboratory also provided the evidence that the effect of genetic variants in TNFSF15 and XAB2 on the susceptibility to gastric cancer could be modified by age (*Gao et al., 2019*; *Pei et al., 2015*). These findings further verified the importance of interaction of age with the genetic polymorphism on the development of gastric cancer.

Gender was also an important factor affecting the development of cancers (*Li et al., 2019*; *Lou et al., 2020*). When stratified by gender, our data shows that individuals with the *C7* rs1061429 AA genotype have an increased risk of gastric cancer in both men and women, but the risk in women is significantly higher than that in men. This might be related to different lifestyles between men and women (*Song et al., 2008*) and other risk factors, such as diet, microbial virulence, and Hp infection (*Cover & Peek Jr, 2013*; *Xia et al., 2016*). Similarly, researchers found that *C7* rs1063499 GG genotype increased the risk of liver cancer among men, but not among women (*De Lima et al., 2018*). Our previous study demonstrated that TNFSF15 -638 GG genotype was associated with an increased risk of SCLC among males compared with the AA genotype, but not among females (*Gao et al., 2019*). These findings further indicated the contribution of gender to cancer development.

There are still some limitations. The sample size is relatively small after stratification. In addition, as one of the best-established environmental factors of gastric cancer, *H. Pylori* infection should be considered in this study (*Wang et al., 2014*). In our future research, we should pay more attention to the exploration of reducing the impact of more environmental compounding factors.

## CONCLUSIONS

In summary, our study demonstrated that *C7* genetic variants were related to gastric cancer's susceptibility which implying the critical role of the complement genes in the development of gastric cancer.

**List of abbreviations**

| | |
|---|---|
| *C7* | Complement component 7 |
| **GC** | Gastric cancer |
| **GTEx** | Genotype-Tissue Expression |
| **GEO** | Gene Expression Omnibus |
| **PCR-RFLP** | Polymerase chain reaction-restriction fragment length polymorphism technique |
| **MAC** | Membrane attack complex |
| **HWE** | Hardy-Weinberg equilibrium |

### Funding

This study was funded by the Foundation of Key Project of Natural Science Foundation of Hebei province of China (H2017209233 to X. Zhang), and the Leader talent cultivation plan of innovation team in Hebei province (No. LJRC001). The funders had no role in study design, data collection and analysis, decision to publish, or preparation of the manuscript.

### Grant Disclosures

The following grant information was disclosed by the authors:
Foundation of Key Project of Natural Science Foundation of Hebei province of China: H2017209233.
Leader talent cultivation plan of innovation team in Hebei province: No. LJRC001.

### Competing Interests

The authors declare there are no competing interests.

### Author Contributions

- Siyue Wang conceived and designed the experiments, performed the experiments, analyzed the data, prepared figures and/or tables, and approved the final draft.
- Wenqian Hu performed the experiments, authored or reviewed drafts of the paper, and approved the final draft.

- Yuning Xie and Zhenxian Jia analyzed the data, prepared figures and/or tables, and approved the final draft.
- Hongjiao Wu performed the experiments, prepared figures and/or tables, and approved the final draft.
- Zhi Zhang conceived and designed the experiments, prepared figures and/or tables, and approved the final draft.
- Xuemei Zhang conceived and designed the experiments, authored or reviewed drafts of the paper, and approved the final draft.

## Human Ethics

The following information was supplied relating to ethical approvals (i.e., approving body and any reference numbers):

North China University of Science and Technology granted Ethical approval to carry out the study within its facilities (2019021).

## Data Availability

Data for C7 rs1061429 and C7 rs1376178 are available in the Supplemental Files.

## Supplemental Information

Supplemental information for this article can be found online at http://dx.doi.org/10.7717/peerj.12816#supplemental-information.

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
