# Peer review of "Functional genetic variants in complement component 7 confer susceptibility to gastric cancer"

_PeerJ, doi:10.7717/peerj.12816_

## Round 0.1 · original submission · Major Revisions

Dear authors:
I have received the comments and concerns rose by leading experts in the field regarding your manuscript and important issues have been detected in the manuscript. I agree with reviewer 1 that the study has potential clinical significance and finding novel markers for cancer is always relevant.

Nevertheless, I kindly request that all comments are properly analyzed and provide a revised version of the manuscript, in particular with the statistical analysis. I also suggest as pointed out by reviewer 1, the title is misleading, the authors do not provide functional analysis and should revise the potential functional experiments to be done.

Reviewer 2 is suggesting an important analysis to be done. Also, please revise the comments of reviewer 2, they are relevant for the validity of the findings.

I hope we can have your revisions soon.
Warm regards

·

Basic reporting

Please see my additional comments.

Experimental design

Please see my additional comments.

Validity of the findings

Please see my additional comments.

Additional comments

This manuscript by Siyue Wang et al. entitled “Functional genetic variants in complement component 7 confer susceptibility to gastric cancer” investigated the relationship between complement component 7 polymorphisms and the risk of gastric cancer (GC). This study is interesting and of potential clinical significance if the following issues are revised.
1. In text line 143, “After adjusted by gender, age, drinking and smoking status, non-conditional logistic regression analysis showed that the distribution of C7 rs1376178 AA and CA genotypes was statistically different between cases and controls (P<0.001)”. Why were these variables, such as gender, age, drinking and smoking status selected?
2. In text line 163 “individuals with rs1061429 AA had an increased risk of gastric cancer among youngers (OR=2.84, 95%CI=1.39-5.80, P=0.004) and non-smokers (OR=2.79, 95%CI=1.63-4.77, P<0.001), but not among elders (OR=1.81, 95%CI=1.00-3.28, P=0.050) and smokers (OR=1.03, 95%CI=0.43-2.5, P=0.947)”. What are the possible explanations?
3. In text line 166, “the stratification analysis by gender or drinking status showed that rs1061429 AA genotype was contributed to the risk of gastric cancer regardless of gender and drinking status (OR = 1.94, 95%CI = 1.13-3.33, P = 0.017 for males; OR=2.87, 95%CI=1.26-6.54, P=0.012 for females; OR=1.92, 95%CI=1.18-3.13, P=0.008 for nondrinkers; OR=5.23, 95%CI=1.34-20.40, P=0.017 for drinkers)”. Isn’t OR=5.23 significantly different from OR=1.92?
4. In text line 176, “we explore the potential function of gene C7 in gastric cancer, based on GEPIA data, our results showed that the level of C7 mRNA in gastric cancer tissues (n=408) was significantly depressed compared with that in adjacent normal tissues (n=211)”. How does the expression of C7 reflect the potential function? Is SNP correlated with C7 expression level? Is C7 level associated to survival in gastric cancer?
5. In Discussion, authors discussed the role of C7 in other types of cancers. Are C7 rs1376178 and rs1061429 SNPs also associated to other cancers?
6. In text line 203, “we speculated that the change of C to A in C7 rs1376178 site enhanced the binding capability of transcription factors STAT1”. In C7 rs1376178 C>A samples, is STAT1 upregulated?
7. In text line 242, “functional experiments are still needed to be done in the future”. What functional experiments? Although the title is “Functional genetic variants in complement component 7 confer susceptibility to gastric cancer”, authors did not provide any functional data in the manuscript.

Reviewer 2 ·

Basic reporting

Comment 1: Line 67-68, it seems a bit abrupt to talk about DAF without first introducing to the readers on DAF and how it's related to Complement component family. Furthermore, 2-3 cases should be given to support "Complement gene polymorphism is closely related to the occurance of cancer"

Comment 2: In the method part, please provide the criteria for people that were selected as smokers and drinkers.

Experimental design

Comment: It is inappropriate to call Fig. 1 a "Function analysis of C7" when it's merely a expression profiling analysis. In 1(A), the authors have studied only on the rs1061429 genotype but not rs1376178, please explain why. This reviewer is also concerned about the imbalanced group sizes in 1(B), a reduced analysis with an equal number of samples per group should be performed to confirm the findings. It is confusing to the readers to say " gastric cancer compared to levels in normal stomach tissues" when the comparison was actually done between the cancerous tissues and adjacent normal tissues, which could have a totally different C7 expression than completely cancer-free tissues. Last but not least, the authors should consider repeating the same analysis in (A) and (B) using your own genotype and gene expression data which were collected on peripheral blood DNA from cancer patients and healthy people rather than gastric tissue DNA from cancerous and adjacent tissues and compare if the findings remain the same.

Validity of the findings

Comment: More analyses should be done to support the valuable role of C7 as a prognostic tool for gastric cancer. The following are some of the reviewer's suggestions:

1. Kaplan-Meier curves according to the C7 expression levels for progression free survival and overall survival of patients with gastric cancer


2. Kaplan-Meier curves according to different variants in each C7 genotye for progression free survival and overall survival of patients with gastric cancer


3. Odd ratios in association of C7 expression level with clinicopathological factors such as histology, grade and clinical stage


4. Odd ratios in association of different C7 variant in each genotype with clinicopathological factors such as histology, grade and clinical stage

Additional comments

None

---

## Round 0.2 · accepted · Accept

Dear authors,

Thank you for submitting a revised version of the manuscript “Functional genetic variants in complement component 7 confer susceptibility to gastric cancer”. After revising the comments by the reviewers and carefully addressing the response letter and new manuscript, I am happy to inform you that the manuscript is suitable for publication in Peer J.

Thank you so much for considering PeerJ for your research.

Best regards and best wishes for 2022.

·

Basic reporting

Authors have successfully responded to all my comments.

Experimental design

None.

Validity of the findings

None.

Additional comments

None.